# Exogenous Glycine Betaine Application Improves Freezing Tolerance of Cabbage (*Brassica oleracea* L.) Leaves

**DOI:** 10.3390/plants10122821

**Published:** 2021-12-20

**Authors:** Kyungwon Min, Yunseo Cho, Eunjeong Kim, Minho Lee, Sang-Ryong Lee

**Affiliations:** 1Department of Biological and Environmental Science, Dongguk University, Seoul 04620, Korea; kwmin@dgu.ac.kr (K.M.); cys9968@dgu.ac.kr (Y.C.); xdmswjdx@dongguk.edu (E.K.); 2Department of Life Science, Dongguk University-Seoul, Goyang 10326, Korea

**Keywords:** compatible solute, freeze-thaw injury, osmolyte, reactive oxygen species, antioxidant capacity, lipid peroxidation

## Abstract

Exogenous glycine betaine (GB) application has been reported to improve plant tolerance to various abiotic stresses, but its effect on freezing tolerance has not been well studied. We investigated the effect of exogenous GB on freezing tolerance of cabbage (*Brassica oleracea* L.) leaves. Seedlings fed with 30 mM GB via sub-irrigation showed effectively assimilated GB as evident by higher GB concentration. Exogenous GB did not retard leaf-growth (fresh weight, dry weight, and leaf area) rather slightly promoted it. Temperature controlled freeze-thaw tests proved GB-fed plants were more freeze-tolerant as indicated by lower electrolyte leakage (i.e., indication of less membrane damage) and alleviating oxidative stress (less accumulation of O_2_^•−^ and H_2_O_2_, as well as of malondialdehyde (MDA)) following a relatively moderate or severe freeze-thaw stress, i.e., −2.5 and −3.5 °C. Improved freezing tolerance induced by exogenous GB application may be associated with accumulation of compatible solute (proline) and antioxidant (glutathione). GB-fed leaves also had higher activity of antioxidant enzymes, catalase (CAT), ascorbate peroxidase (APX), and superoxide dismutase (SOD). These changes, together, may improve freezing tolerance through membrane protection from freeze-desiccation and alleviation of freeze-induced oxidative stress.

## 1. Introduction

While plants are highly tolerant to sub-zero temperatures during the dormancy in winter, the sensitive tissues of plants, including buds, new leaves, and flowers, are typically vulnerable to frost during the active growth [1]. Accordingly, the unseasonal frosts, especially during the spring, can negatively affect growth and reproduction of plants resulting in economic losses [2,3,4,5]. Under a natural frost episode, ice-nucleation is typically initiated around the extracellular spaces, in part, due to extracellular fluid being at higher freezing point than the intracellular fluid. Extracellular ice formation causes a decrease in water potential outside the cell since the water potential of ice is much lower than liquid water at a given temperature. Consequently, water potential gradient decreases driving the water-efflux from cell to extracellular ice, hence, plant cell can experience freeze-desiccation. After thaw-rehydration, freeze-thaw stressed tissues exhibit various cellular dysfunctions (Figure 1). By far, two of such symptoms have been predominantly explored to explain the mechanism of freeze-thaw injury: (1) physico-molecular perturbations in plasma membrane, as evident by electrolyte leakage, and (2) cellular accumulation of reactive oxygen species (ROS), causing oxidative injury to macromolecules [1,6,7,8] (Figure 1). Therefore, reduction of electrolyte leakage and detoxification of accumulated ROS could be among the main strategies for improving freezing tolerance (FT) of plants.

Plants from temperate regions can improve their FT upon exposure to cold temperature through a process called cold acclimation [1,9,10]. From that perspective, cold acclimation is a very complicated process involving various physiological and biochemical changes, including reduced growth/water content, increased antioxidants, hormone level, and accumulation of compatible solutes/osmolytes [11,12,13]. This proposes that exogenous application of compatible solute/osmolyte could potentially be one of the intervention strategies to improve plants’ FT. Glycine betaine (GB) is one of such osmolytes that play an important role in cellular osmotic adjustment, which is associated with improved freezing tolerance through increase in resistance to freeze-induced dehydration/desiccation [14,15]. GB has been also known to function as a compatible solute that effectively stabilizes the quaternary structures of enzymes/proteins and maintains a highly ordered state of membranes [14,16,17]. Accordingly, several studies have investigated its effect on stress tolerance and found that exogenous GB application improved plant tolerance against various abiotic stresses, including salt [18], chilling [19], and drought [20].

Spring cabbage (*Brassica oleracea* L.) is an economically valuable horticultural crop in Asia, particularly in Korea, due to its high consumption [21]. Though somewhat freeze-tolerant [22], it is still sensitive to unseasonal spring frost, leading to frost-damage and an economic loss [23]. The occurrence of such sudden/erratic spring frost is predicted to increase in future of East Asia due to vagaries of climate change [3]. Therefore, development of strategies to enhance FT would be favorable to cabbage farming industry. The main goal of the present study was to determine whether exogenous GB application could alleviate freeze-thaw related deterioration of cabbage leaves, as quantified by various physiological parameters, i.e., leaf-growth measurement, leaf concentration of GB, electrolyte leakage, malondialdehyde (MDA) accumulation (an indicator of lipid peroxidation), activity of antioxidant enzymes, histochemical detection of ROS, and proline content analysis.

## 2. Results

### 2.1. Effect of Exogenous GB on Leaf-Growth

Leaf-area of seedlings treated with GB was larger than non-GB fed control (hereon referred as ‘NGB’) by 4.5%; DW/FW of GB-fed leaves was small but significantly higher than NGB (Table 1).

Since DW/FW between NGB and GB-fed leaves was statistically different, all biochemical analyses were hereafter quantified on DW basis.

### 2.2. LT_50_ of Cabbage Leaves

A sigmoid curve in response to freeze-thaw stress (−1 to −10 °C) for ‘Myeong-sung’ cabbage leaves is presented in Figure 2; freeze-thaw injury at each individual treatment temperature was estimated by electrolyte-leakage based assay. Minimum and maximum injury percentages were ~0.7% and ~90.7%, respectively, with −2.9 °C corresponding to LT_50_, the temperature causing a 50%injury or plant kill of cabbage leaves, i.e., ~46%.

### 2.3. Effect of Exogenous GB on FT and GB Concentration

Four test freezing temperatures, i.e., −2.0, −2.5, −3.0, and −3.5 °C, were first selected to compare FT of NGB and GB-fed leaves. Leaves fed with GB were significantly more freeze-tolerant than NGB as evident by a reduction in injury percentages by 72.9, 50.1, 41.9, and 28.5% at −2.0, −2.5, −3.0, and −3.5 °C, respectively (Figure 3A). Based on LT_50_ of −2.9 °C (Figure 2), the two test temperatures, i.e., −2.5 and −3.5 °C, were selected for further studies to clarify the cellular mechanisms of GB-induced FT since they, respectively, represented moderate and relatively severe injury level.

GB concentration in GB-fed leaves was higher than NGB by ~10% (Figure 3B), indicating that seedlings effectively absorbed and assimilated GB.

### 2.4. MDA Concentration

GB-fed leaves had ~44 and ~52% less MDA concentration following exposure to freeze-thaw stress at −2.5 and −3.5 °C, respectively, compared to NGB (Figure 4).

### 2.5. ROS (O_2_^•−^ and H_2_O_2_) Staining 

The distribution of superoxide (O_2_^•−^) (blue stain) and hydrogen peroxide (H_2_O_2_) (brown stain) in UFC and the freeze-thaw stressed leaves is shown in Figure 5. A higher accumulation of O_2_^•−^ and H_2_O_2_ in stressed tissues compared to UFC was apparent across all treatments, as evident by higher intensities of O_2_^•−^ (Figure 5A,B) and H_2_O_2_ (Figure 5C,D), respectively. Moreover, ROS accumulation increased with severity in freezing stress (−2.5 versus −3.5 °C). Staining intensities for the two ROS particles were visually lower in GB-fed leaves compared to NGB after exposure to −2.5 and −3.5 °C.

### 2.6. Antioxidant Enzyme Activities and Leaf-GSH Concentration

Data of CAT, APX, and SOD activities in UFC, as well as freeze-thaw stressed leaves, are presented in Figure 6. All enzyme activities across all treatments were repressed following freeze-thaw stress at −2.5 and −3.5 °C relative to corresponding UFCs. Furthermore, all enzyme activities were lower at −3.5 °C than at −2.5 °C across all treatments. However, CAT, APX, and SOD activities, respectively, in GB-fed leaves were ~32.5, ~19.3, and ~9.2% higher than NGB following freezing at −2.5 °C, and were ~35.7, ~37.7, and ~56.7% higher than NGB after freezing at −3.5 °C. Leaf-GSH concentration in GB-fed leaves was ~11% higher than NGB (Figure 6D).

### 2.7. Proline Concentration

Proline concentration of GB-fed leaves was significantly higher (~3.1-fold) than that of NGB (Figure 7).

### 2.8. Correlation Analysis of Physiological and Biochemical Traits

Pearson’s correlation analysis was performed to evaluate the relationships between measured physiological and biochemical parameters of cabbage leaves categorized into four different experimental conditions (Figure 8). In terms of samples grown at ambient condition, significant positive correlations were exhibited for all parameters between GB, GSH, and proline (Figure 8A). For UFC samples, there was a significant positive relationship between electrolyte-leakage and SOD (Figure 8B). For tissues stressed at −2.5 °C, significant correlations were observed among MDA, electrolyte-leakage, CAT, APX, and SOD—positive correlation (i.e., electrolyte-leakage versus MDA, CAT versus APX, and CAT versus SOD, as well as APX versus SOD) (Figure 8C). Regarding the tissues stressed at −3.5 °C, significant correlations, as in the case of −2.5 °C, was observed between electrolyte, MDA, CAT, APX, and SOD—positive correlation (i.e., MDA versus electrolyte-leakage, CAT versus APX, and CAT versus SOD, as well as APX versus SOD) (Figure 8D).

## 3. Discussion

Numerous studies have reported that exogenous GB application improved plant tolerance against various abiotic stresses, including heat, drought, salt, and chilling [14,18,19,20,24]. However, data on its effect of plant FT are deficient. Although a few other studies have reported on the effect of exogenous GB on FT, such as in wheat/barley [25], alfalfa [26], isolated spinach thylakoids [27], strawberry [28], and *Arabidopsis thaliana* [29], the present study was the first to explore the hypothesis that exogenous GB application mitigates freeze-thaw injury to cabbage leaf tissues, as quantified by various physiological parameters. Moreover, our research, as different from the first three studies, evaluated FT using a temperature-controlled freeze-thaw protocol, regulating gradual and realistic cooling and thawing with ice-nucleation. In the present study, we provided evidence that 30 mM exogenous GB enhances FT (reduced membrane injury and lower ROS accumulation) of cabbage leaves without causing any negative effect on leaf health.

### 3.1. GB Application and Leaf-Growth

The effect of GB on plant growth, as well as stress tolerance, relies on the mode of application and concentrations [14,30]. Accordingly, in preliminary experiments, we tested the effect of five GB concentrations, i.e., 5, 10, 20, 30, and 40 mM, on FT as sub-irrigation treatments. Seedlings fed with 40 mM GB exhibited somewhat stagnant growth compared to NGB, whereas the other four concentrations did not show any detrimental effect on growth, of which only the 30 mM GB was selected for further experiments; this was because 30 mM GB-fed leaves were the most freezing tolerant compared to the other three concentrations (Appendix A). Higher leaf GB concentration in GB-fed leaves than NGB (Figure 3B) indicates that seedlings effectively took GB up.

Several studies noted that GB application promoted plant growth at optimal, as well as stressed, conditions, such as in ryegrass (20 or 50 mM GB) [31], pea (4 mM GB) [32], snap bean (5 mM GB) [33], and rice (5 or 10 mM GB) [34]. Our results of 5% bigger leaf area in GB-fed leaves than in NGB (Table 1) support their conclusions. Improvement in GB-induced leaf-growth may be associated with enhanced photosynthetic rates, as well as the stabilizing effect, of GB on the oxygen-evolving activity (i.e., an evolution of oxygen from oxidizing water) via protection of oxygen-evolving complex in PSII against dissociation of extrinsic proteins and Mn cluster [17]. Indeed, snap bean leaves treated with GB had relatively higher levels of photo-assimilates under optimal conditions [33]. Moreover, GB application itself enhanced photosynthetic CO_2_-assimilation rates of maize under non-stress conditions [35].

### 3.2. GB Application Improves FT by Mitigating Membrane Damage

GB-fed leaves were significantly less injured compared to NGB at all stress levels based on electrolyte leakage assay (Figure 3A). This result indicates that GB application improves FT of cabbage leaves by reducing cellular membrane damage caused by freeze-thaw stress. According to Reference [25], GB, acting as an osmolyte, enhances resistance against osmotic stress through increase in the osmolality of the cell, and their study showed increase in leaf osmolality by external GB under ambient conditions. Although the osmolality was not measured in the present study, it may be reasonable that GB-fed leaves have higher resistance against freeze-induced desiccation stress than NGB due to GB-induced higher osmolality. This assumption may be partially supported by our data that GB-fed leaves had significantly higher leaf-GB concentration than NGB (Figure 3B).

The increase in electrolyte leakage in freeze-thaw injured tissues is associated with compromised membrane transport functions. For instance, three studies have shown a reduced H^+^-ATPase activity concomitant with increased electrolyte leakage in freeze-thaw injured tissues (i.e., onion bulb scale tissues, mesophyll cell of pine needles, and cellular membrane of *Helianthus tuberosus*) [36,37,38]. Furthermore, recovery of H^+^-ATPase activity during post-thaw periods leads to the re-uptake of leaked ions [36]. It has been noted that GB, acting as zwitterion, can interact with hydrophilic and/or hydrophobic domains of membranes, which can contribute to stabilizing and maintaining its functional integrity [39,40,41]. For example, tomato plants transformed with the choline oxidase gene *9osi*, responsible for oxidation of choline to GB, had higher enzymatic activity of plasma membrane H^+^-ATPase under nutritional stress [42]. In light of the above discussion, data from the present study suggests that exogenous GB application not only alleviates suffering from freeze-induced desiccation stress (i.e., acting as osmolyte for osmotic adjustment) but also constitutes favorable conditions for H^+^-ATPase function, which together alleviates freeze-induced membrane damage, as evident by lower electrolyte leakage. Others have also reported similar results of lower electrolyte leakage via exogenous GB against freezing [25,26,29], chilling [19], salt [31], and drought [43].

Induction of FT (as in the case of cold acclimation) generally relates to decrease in cellular hydration status [13]. Our data also support this notion in that GB-fed leaves, i.e., marginally less hydrated as compared to NGB (Table 1), had higher FT. In order to further explore possible physiological/biochemical explanation for GB-induced FT, two stress temperatures (i.e., −2.5 and −3.5 °C) were selected based on the freeze-response curve of NGB (Figure 2); these two test temperatures represented physiologically relevant freeze-thaw injury levels: relatively moderate level (~28%) and a level more lethal than LT_50_ (~60%) of the maximum injury.

### 3.3. GB Application Enhances Antioxidant System

Plant tissues exposed to freeze-thaw stress accumulate excessive ROS (e.g., O_2_^•−^ and H_2_O_2_) which, if not properly removed by antioxidants, cause oxidative damages to various cellular components, including cellular membrane [8,44]. Our data are also consistent with these reports since freeze-thaw stressed leaves showed higher amount of O_2_^•−^ and H_2_O_2_ (visual intensity) compared to UFC (Figure 5). Similar visual observation was reported by References [45,46] wherein spinach leaves subjected to freeze-thaw stress had greater amount of O_2_^•−^ and H_2_O_2_ than non-stressed leaves. More importantly, ROS accumulation was lower in GB-fed leaves than in NGB at both stress levels (Figure 5), suggesting a possibly enhanced antioxidant capacity of GB-fed leaves.

Indeed, our data support this notion by showing a close correspondence between ROS abundance and antioxidant enzyme activities. For example, GB-fed leaves had higher CAT and APX activities than NGB at both stress levels (Figure 6A,B), which might help more efficiently remove excessive H_2_O_2_ (Figure 5C,D). The positive correlation between CAT and APX under freeze-thaw stress can support this observation (Figure 8C,D). SOD activity in GB-fed leaves was also higher than NGB at both stress level (Figure 6C), which might be responsible for less O_2_^•−^ accumulation (Figure 5A,B). A higher APX activity in GB-fed leaves can be supported by a higher leaf-GSH concentration (Figure 6D) since GSH works in concert with APX via ascorbate-glutathione cycle [47,48].

Oxidative injury caused by excessive ROS accumulation could manifest at the cellular membrane by accumulation of MDA (i.e., a biomarker for lipid peroxidation), ultimately leading to increase in electrolyte leakage in freeze-thaw tissues [1,7]. Therefore, a lower MDA concentration, as well as electrolyte leakage, in GB-fed leaves than NGB at both stress levels (Figure 2 and Figure 4), along with their positive correlation at freeze-thaw stressed tissues (Figure 8C,D), further supports enhanced antioxidant capacity by exogenous GB application. Numerous studies have also reported that exogenous GB application enhanced activity of antioxidant enzymes, along with reduction of MDA concentration, against various abiotic stresses [31,49,50,51].

Improved antioxidant capacity by exogenous GB application may be mediated by the Ca^2+^ sensor proteins, including the Ca^2+^-dependent protein kinase (CDPKs) and calcineurin B-like proteins (CBLs); these proteins are crucial for enhanced antioxidant enzyme activities in plants [13]. A transcriptomics study performed by Reference [52] supported this assumption that GB-treated maize seedlings under salt stress had higher tissue Ca^2+^ content and upregulated three protein kinase-related genes, including putative CDPK, CBL-interacting protein kinase, and CBL-interacting protein kinase 14-like, which accompanied upregulation of antioxidant-related genes, such as peroxidase 67, APX1-cytosolic ascorbate peroxidase, glutathione transferase 9, etc. Though transcriptomic activity of these was not directly measured in the present study, it is tempting to speculate that exogenous GB application might have led to increased tissue Ca^2+^ concentration, upregulated Ca^2+^ signaling related genes, and, thus, increased antioxidant enzymatic activities.

### 3.4. GB Application Accumulates Proline

Proline, a compatible solute, has been widely noted to accumulate under stress conditions with a vital role in cellular osmotic adjustment, membrane/protein stabilization and ROS scavenging [53]. In the present study, the freeze-tolerant GB-fed cabbage leaves had higher level of proline (~3.1-fold) than NGB (Figure 7). In support of our results, others have also reported exogenous GB-induced proline accumulation and concomitant alleviation of injury induced by abiotic stresses, including chilling in zucchini/pears [54,55] and drought in *Axonopus compressus* [43]. No experiment was performed in our study to understand why exogenous GB application increases proline accumulation. However, two studies provide such a potential explanation where GB-treated fruits showed increased activities of two key enzymes for proline biosynthesis, i.e., Δ1-pyrroline-5-carboxylate synthetase (P5CS) and ornithine d-aminotransferase (OAT) in GB-fed fruits, under chilling conditions (0–1 °C) [54,55]; both studies also showed decreased activity of proline dehydrogenase (PDH). Conceivably, higher levels of proline in GB-fed leaves growing at ambient conditions in our data, including their positive correlation (Figure 8A), could potentially reflect higher activities of P5CS or OAT with decreased activity of PDH, a proposal warranting further investigation.

## 4. Materials and Methods

### 4.1. Plant Material

Seeds of *Brassica oleracea* L. cv. ‘Myeong-Sung’ (Kyoungshin seeds, Inc., Kyungbuk, Korea) were sown on plug flats filled with cultural media (Heuksalim Lab., Chungbuk, Korea) and germinated in growth chamber at 20/18 °C (D/N) under average photosynthetically active radiation (PAR) of ~300 μmol m^−2^ s^−1^ with 12-h photoperiod. Seedlings were watered as needed via sub-irrigation (~3-d interval). Fourteen-days-old cabbage seedlings were sub-irrigated with 30 mM GB dissolved in tap water or only with tap water (i.e., non-GB fed control; NGB); this means that seedlings were treated with GB only once. Twenty-one-day-old cabbage seedlings (i.e., 7 days after GB application) were employed for experiments as described below.

### 4.2. Leaf Growth Measurement

Leaf growth was estimated by measuring fresh weight (FW), dry weight (DW), and leaf area between NGB versus GB-fed leaves. Briefly, 10 pairs of first true two leaves (total 20 leaves) per treatment were excised from seedlings and then, used to measure FW. Subsequently, leaf-area was measured using LI-3100 Area Meter (LI-COR, Inc., Lincoln, NE, USA), followed by DW measurement after oven-drying leaves at 75 ± 1 °C for 96 h.

### 4.3. Freezing Tolerance (FT) Determination

FT of cabbage-leaf was determined based on the electrolyte leakage-based laboratory freeze-thaw protocol [7]. Briefly, the first two true petiolate leaves excised from a seedling were placed in a 2.5 cm × 20 cm test tube containing 150 μL deionized water and then slowly cooled in a glycol bath (JSCR-30C; JS Research Inc., Gongju, Chung-Nam, S. Korea) to various freezing temperatures at −1 °C·h^−1^ following ice-nucleation at −1 °C; ice-nucleation was conducted by dropping an ice-chip into each test tube. Tissues were kept for 30 min at each test temperature and thawed on ice overnight. Unfrozen control (UFC) leaves were kept at 0 °C throughout the freeze-thaw cycle. The next morning, samples were taken out of the ice and gradually thawed at 4 °C for 1 h, followed by 1 h at room temperature (~20 °C). Subsequently, 20 mL of deionized water was added to each test tube, and samples were subjected to vacuum infiltration for three times, each with 3 min, and shaken for 1.5-h at 250 rpm. First, electrolyte leakage was determined for each sample using a conductivity meter. Second, electrolyte-leakage was also measured at room temperature for each autoclaved sample. Percent injury, indicating the FT of samples under each test temperature, was calculated from the data of electrolyte leakage according to Reference [56].

In order to select the freezing treatment temperatures, firstly, a LT_50_ curve was plotted for NGB samples by freezing the leaves from −1 to −10 °C; tubes were removed from the glycol bath at −1 °C intervals. This experiment was independently repeated thrice, each with 5 technical replicate/temperature. A sigmoid curve fitting the Gompertz function was plotted using percent injury at individual treatment temperature from these experiments [56]; a LT_50_, mid-point between the minimum and maximum injury, was interpreted as the leaf FT. Based on this sigmoid curve, four freezing treatment temperatures, i.e., −2.0, −2.5, −3.0, and −3.5 °C, were first determined for comparing FT of NGB versus GB-fed leaves.

### 4.4. Determination of GB Concentration

Leaf tissues excised from NGB and GB-fed plants were used to determine the GB concentration, as detailed by Reference [57]. Briefly, the finely ground frozen tissue (10 mg) was mixed with 1.5 mL of 2N H_2_SO_4_, followed by heating up in water bath (60 °C) for 10 min. The extracts were then centrifuged at 10,000× *g* for 25 min at room temperature (~20 °C). The supernatant (125 μL) was mixed with cold KI-I_2_ (prepared with 15.7 g of iodine and 20 g of KI in 100 mL of sterilized water). The samples were then kept at 0 to 4 °C for 16 h followed by centrifuged at 10,000× *g* for 30 min at 0 °C. The supernatant was gently removed, and then GB crystals were dissolved with 1.4 mL of 1,2-dichloroethane for 48 h at room temperature (~20 °C) before being read at 290 nm using 1,2-dichloroethane as a blank. The GB concentration was determined from a standard curve (0, 50, 100, 150, 200, 250, 300, and 350 μg ml^−1^, i.e., equivalent to ~2990 μM), and the concentration values were then calculated based on DW.

### 4.5. Determination of Malondialdehyde (MDA) Concentration

Leaves from NGB and GB-fed plants that were exposed to freeze-thaw stress at −2.5 and −3.5 °C and their corresponding UFCs were used to measure MDA concentration, as described in Reference [7]. Briefly, samples were ground into fine power using liquid nitrogen, and 100 mg tissue per temperature per treatment was homogenized with 1.5 mL cold 10% trichloroacetic acid. Samples were then vigorously vortexed before centrifugation (10,000× *g*) for 20 min at 4 °C. The supernatants were allocated into three technical replicates of 400 μL and each mixed with 400 μL of 0.5% 2-thiobarbituric acid. Mixtures were then heated at 95 °C for 40 min, followed by cooling on ice for 10 min and centrifugation (10,000× *g*) for 10 min at room temperature (~20 °C). The absorbance of supernatant was measured at 450, 532, and 600 nm employing a spectrophotometer. The MDA concentration was estimated by using the formula: [MDA] = 6.45 × (*A*_532_ − *A*_600_) − 0.56 × *A*_450_ [58]. The concentration values were converted to nmoles per g. DW.

### 4.6. ROS Staining

Superoxide (O_2_^•−^) and hydrogen peroxide (H_2_O_2_) distribution were visualized using stain by nitroblue tetrazolium (NBT) or 3,3′-diaminobenzidine (DAB) method, respectively, as detailed by Reference [59]. This experiment was independently conducted twice, each with four replicates (2 leaves per replicate) per temperature per treatment. A representative picture showing staining intensities is presented.

### 4.7. Measurement of Antioxidant Enzyme Activities

The activities of three antioxidant enzymes, i.e., catalase (CAT), ascorbate peroxidase (APX), and superoxide dismutase (SOD), were measured, as detailed by Reference [58]. Briefly, ground frozen leaf tissues (150 mg) were mixed with 1 mL of 100 mM potassium phosphate buffer (pH 7.0), followed by centrifugation (10,000× *g*) for 25 min at 4 °C. The resultant supernatants were then employed as the enzyme extract for CAT, APX, and SOD. The protein amount in the total enzyme extracts from 150 mg tissues was estimated as detailed by Reference [60].

For CAT activity, the reaction mixture consisted of: 100 mM potassium phosphate buffer (pH 7.0), 20 μL of enzyme extract, and 50 mM H_2_O_2_. One unit of CAT activity was defined as the degradation of 1 μM H_2_O_2_ in 1 min at 240 nm. The reaction mixture for APX activity was composed of: 100 mM potassium phosphate buffer (pH 7.0), 0.5 mM ascorbic acid, 0.1 mM EDTA, 50 μL enzyme extract, and 0.1 mM H_2_O_2_. One unit of APX activity was defined as the conversion of ascorbic acid into monohydroascorbate in 1 min at 290 nm.

The reaction mixture for SOD activity included: 50 mM potassium phosphate buffer (pH 7.8), 15 mM nitro blue tetrazolium, 130 mM methionine, 20 μM riboflavin, 5 μM EDTA, and 50 μL enzyme extract. One unit of SOD activity was defined as the amount of enzyme required to reach 50% inhibition of formazan formation at 560 nm.

### 4.8. Determination of Glutathione (GSH) Concentration

Leaves excised from NGB and GB-fed plants were used to measure the GSH concentration using HPLC, as described by Reference [45]. The finely ground frozen tissues (~0.2 g) were thoroughly homogenized with extraction buffer containing 0.1% trifluoroacetic acid and 200 mM dithiothreitol. The homogenate was centrifuged at 12,000× *g* for 10 min, and resultant supernatant was further centrifuged for 5 min. The sample was then injected into a Spherisorb 5 um ODS column (250 mm × 4.6 mm) for HPLC coupled to 1200 series evaporative light scattering detector (Agilent Technologies, Santa Clara, CA, USA).

### 4.9. Determination of Proline Concentration

Leaves excised from NGB and GB-fed plants were used to determine the proline concentration, as described by Reference [61]. The finely ground frozen tissues (100 mg) were vigorously mixed with 1 mL of 3% sulfosalicylic acid and then heated at 80 °C for 15 min. Samples were centrifuged at 12,000× *g* at room temperature (~20 °C) for 20 min. Subsequently, the supernatant (500 μL) was diluted with 500 uL of distilled water, followed by mixing with 1 mL of ninhydrin and 1 mL of glacial acetic acid. The reaction mixtures were heated in a water bath at 95 °C for 1 h and then allowed to cool in an ice bath before being vigorously mixed with 4 mL of toluene. Samples were then set aside to allow separation of the non-polar and polar phase. The non-polar phase containing the chromophore was used to read absorbance at 520 nm using toluene as a blank. Proline concentration was determined from a standard curve (0.25, 0.5, 1.0, 2.0, and 5.0 mM), and concentration values were then calculated based on DW.

### 4.10. Statistical Analysis

All biochemical analyses were independently repeated thrice, each with 3 technical replicates per treatment. The data were analyzed using Student’s *t*-test (α = 0.05) to estimate any statistically significant differences between two treatments. The pair-wise differences among more than three treatments were compared via the Fisher’s LSD test (α = 0.05). Pearson correlation analysis between the measured physiological and biochemical indices was performed using R software (version 4.1.2, The R Foundation for Statistical Computing, ISBN 3-9000051-07-0).

## 5. Conclusions

In the present study, we first determined the FT of Myeong-sung cabbage leaves as the LT_50_ (i.e., ~−2.9 °C) and then explored the hypothesis that exogenous GB application enhances FT of cabbage leaves. Our study offers evidence that exogenous GB application can improve FT of cabbage leaves by reducing membrane leakage and lowering MDA and ROS accumulation without causing any detrimental effect on leaf growth. Our data indicate that GB-induced FT may be linked to higher accumulation of compatible solutes (i.e., GB and proline) and increased activity of antioxidant enzymes (i.e., CAT, APX, and SOD), as well as non-enzymatic antioxidant (i.e., GSH). Future studies should analyze the effect of exogenous GB application on FT at whole-plant level. Furthermore, additional mechanistic elucidation of GB-induced FT of cabbage leaves is warranted, involving comparative gene expression analyses and monitoring other factors (e.g., specific metabolites or genes induced by GB application) contributing to FT.

## Figures and Tables

**Figure 1 plants-10-02821-f001:**
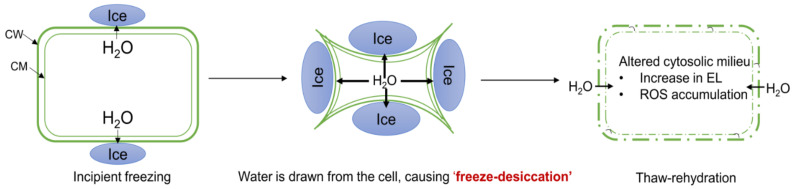
A model illustrating potential cellular event during a freeze-thaw stress. CW, cell wall; CM, cell membrane; EL, electrolyte leakage; ROS, reactive oxygen species.

**Figure 2 plants-10-02821-f002:**
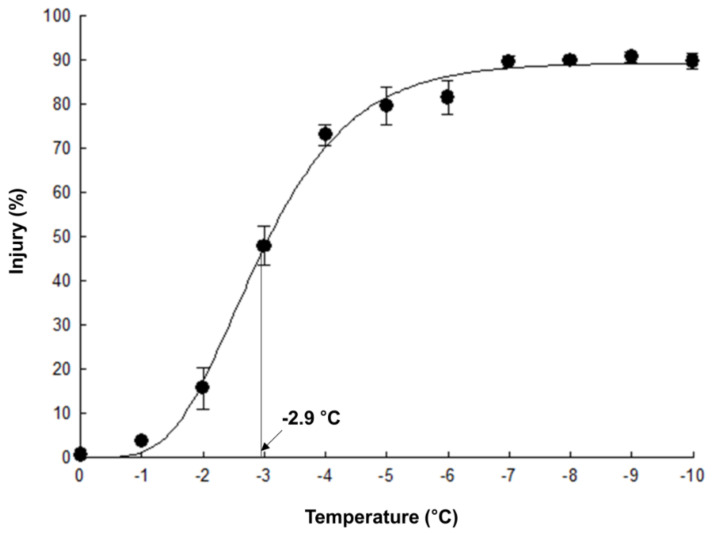
Sigmoid curve of freeze-thaw injury in the leaves of 3-week-old cabbage (*Brassica oleracea* L. cv. Myeong-Sung) seedlings, generated from percent injury at individual treatment temperature employing Gompertz function: LT_50_, a mid-point (46.0% corresponding to −2.9 °C) between the minimum (0.7%) and maximum (90.7%) injury is defined as the temperature causing 50% injury.

**Figure 3 plants-10-02821-f003:**
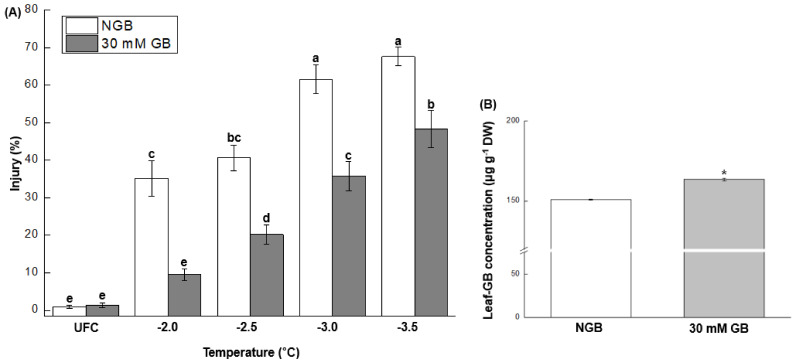
(**A**) Effect of exogenous GB on leaf freezing tolerance of 3-week-old cabbage (*Brassica oleracea* L. cv. Myeong-Sung) seedlings sub-irrigated with water only (non-GB fed control; NGB) and water + 30 mM GB (30 mM GB); injury percent (means ± S.E.) assessed by electrolyte leakage from excised-leaves subjected to freeze-thaw stress at −2.0, −2.5, −3.0, and −3.5 °C. Unfrozen leaves corresponding to each treatment were employed as control (UFC). Different letters indicate significant differences between treatment, analyzed by a Fisher’s LSD test (α = 0.05). (**B**) Leaf GB concentration (means ± S.E.) for NGB and 30 mM GB before exposure to freezing treatments. *, *p* ≤ 0.05, analyzed by Students *t*-test.

**Figure 4 plants-10-02821-f004:**
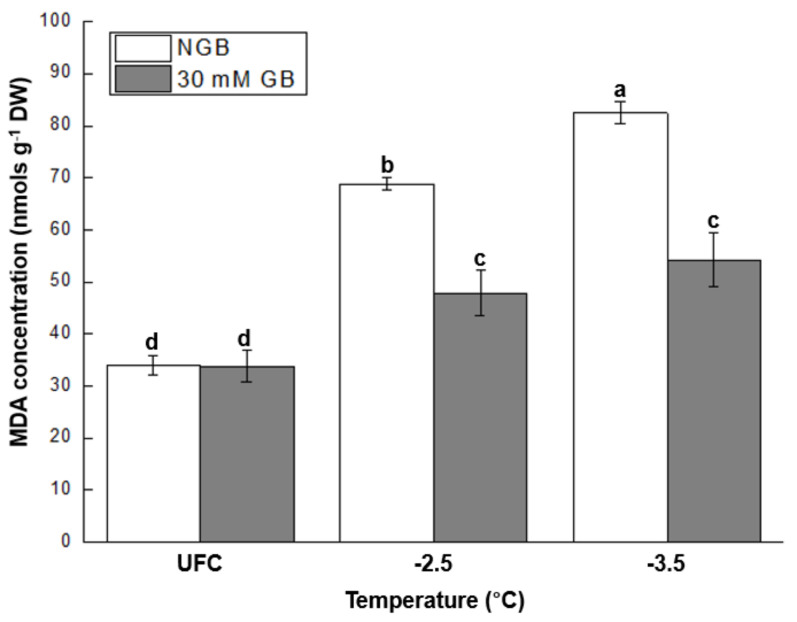
Malondialdehyde (MDA) concentration in unfrozen controls (UFCs) and freeze-thaw stressed cabbage (*Brassica oleracea* L. cv. Myeong-Sung) leaves that were sub-irrigated with water only (non-GB fed control; NGB) and water + 30 mM GB (30 mM GB) before exposure to freeze-thaw stress at −2.5 and −3.5 °C. Different letters indicate significant differences between treatments analyzed by a Fisher’s LSD test (α = 0.05).

**Figure 5 plants-10-02821-f005:**
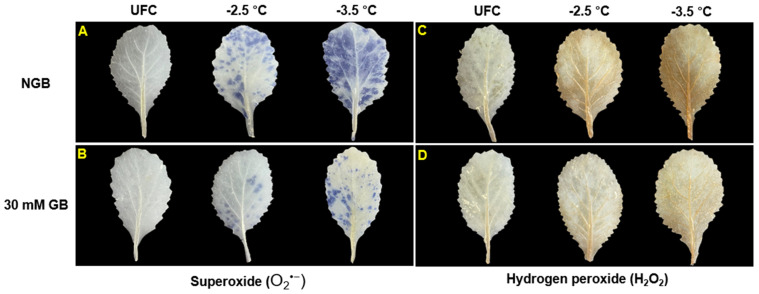
Distribution of superoxide (O_2_^•−^) (**A**,**B**) and hydrogen peroxide (H_2_O_2_) (**C**,**D**) in unfrozen controls (UFCs) and freeze-thaw stressed cabbage (*Brassica oleracea* L. cv. Myeong-Sung) leaves that were sub-irrigated with water only (non-GB fed control; NGB) and water + 30 mM GB (30 mM GB) before exposure to freeze-thaw stress at -2.5 and −3.5 °C.

**Figure 6 plants-10-02821-f006:**
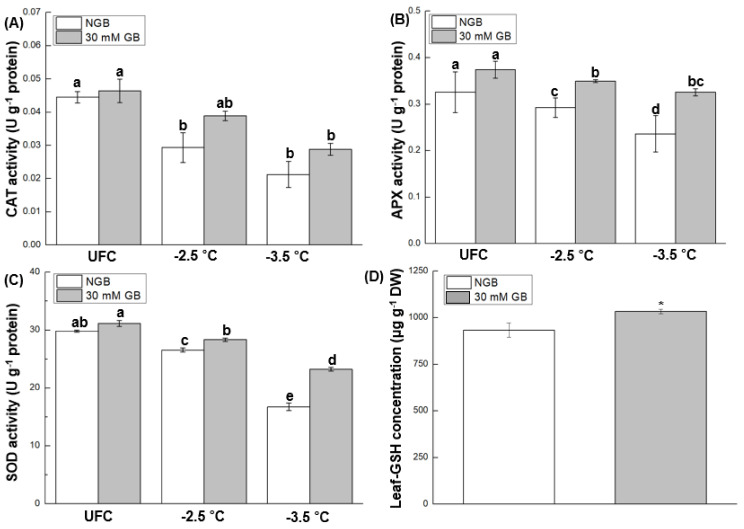
The activity of CAT, APX, and SOD (**A**–**C**) in unfrozen controls (UFCs) and freeze-thaw stressed cabbage (*Brassica oleracea* L. cv. Myeong-Sung) leaves that were sub-irrigated with water only (non-GB fed control; NGB) and water + 30 mM GB (30 mM GB) before exposure to freeze-thaw stress at −2.5 and −3.5 °C; different letters indicate significant differences between treatments analyzed by a Fisher’s LSD test (α = 0.05). (**D**) Leaf-glutathione concentration of cabbage leaves in NGB and 30 mM GB before exposure to freezing treatments; values represent the average ± S.E. from three independent experiment, each with 2 replications per treatment. *, *p* < 0.05, analyzed by Students *t*-test.

**Figure 7 plants-10-02821-f007:**
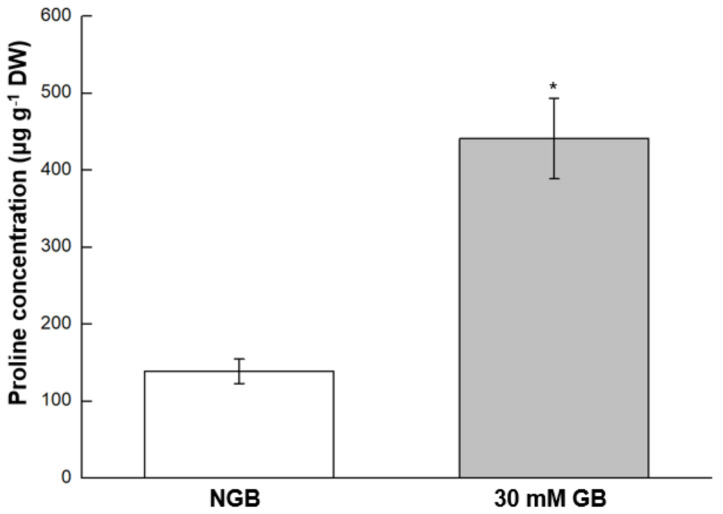
Proline concentration of 3-week-old cabbage (*Brassica oleracea* L. cv. Myeong-Sung) seedlings sub-irrigated with water only (non-GB fed control; NGB) and water + 30 mM GB (30 mM GB) before exposure to freezing treatments. *, *p* < 0.05, analyzed by Students *t*-test.

**Figure 8 plants-10-02821-f008:**
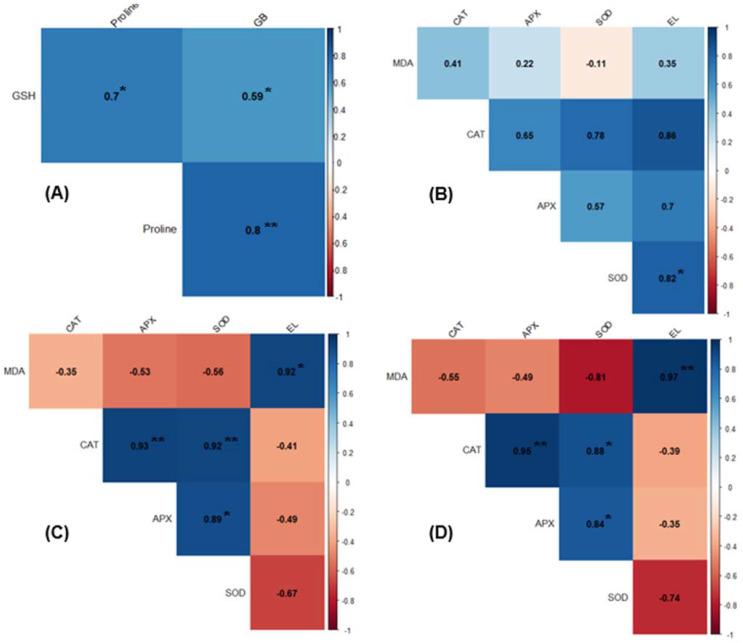
Pearson’s correlation analysis of the physiological/biochemical parameters measured in cabbage (*Brassica oleracea* L. cv. Myeong-Sung) leaves subjected to four different experimental conditions as follows: (**A**) plants grown at ambient condition (20/18 °C; D/N), (**B**) unfrozen control (as called UFC) samples, i.e., kept at 0 °C throughout freeze-thaw cycle, (**C**) plants subjected to freeze-thaw stress either at −2.5 °C or (**D**) −3.5 °C; see materials and methods for specific experimental conditions of four different treatments. GB, glycine betaine; GSH, glutathione, CAT, catalase, APX, ascorbate peroxidase, SOD, superoxide dismutase, MDA, malondialdehyde, EL, electrolyte-leakage. * and ** denote statistical significance at the *p* < 0.05 and 0.01 levels, respectively.

**Table 1 plants-10-02821-t001:** Leaf growth parameters of cabbage seedlings (*Brassica oleracea* L. cv. Myeong-Sung) sub-irrigated with only tap water (non-GB fed control; NGB) and 30 mM GB + tap water (30 mM GB). FW, fresh weight; DW, dry weight.

Growth Parameters	Treatment	
	NGB	30 mM GB
DW/FW ^y^	0.06 ± 0.002	0.07 ± 0.002 ^z^
Leaf area (cm^2^) ^y^	8.8 ± 0.6	9.2 ±0.7 ^z^

^y^ Pooled means ± SE from three biological replications, each including 10 plants. Two leaves per plant were employed, resulting in a total 20 leaves per biological replications, analyzed by Student *t*-test. ^z^
*p* < 0.05.

## Data Availability

The data presented in this paper are available on request from the corresponding authors. The data are not publicly available due to privacy concern.

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
