# Peer review of "Exogenous Glycine Betaine Application Improves Freezing Tolerance of Cabbage (Brassica oleracea L.) Leaves"

_plants, 2021, doi:10.3390/plants10122821_

Round 1

Reviewer 1 Report

The authors studied the general biological laws of the osmolytic action on one cabbage variety. However, the paper lacks information on the justification for the choice of this particular variety. For example, was it selected for a trait of cold or frost tolerance?

Author Response

Dear Mr. Reviewer #1,

We truly appreciate you for taking your time to handle our manuscript. We believe that the comments raised by the reviewer #1 significantly enhance the quality of our study. In addition, we would like to thank positive evaluation on our manuscript. All valuable comments have been fully addressed in our revision. Please refer the uploaded file (Review Response (#1)-Plants-1451053.docx).

Sincerely,

Sang-Ryong Lee

Reviewer 2 Report

Dear Authors,

Your study of GB effects on FT in cabbage was extended for my peer-review. In it, you apply a score well-established and proper methods to assess the physiological mechanisms behind this phenomenon in the economically important crop. At this stage, I suggest some important changes to hopefully increase the strength and impact of your report. Please find these identified by line number, with a flag [MAJOR] wherever a particularly important issue, in my opinion, was found.

32 significant [How to test for significance in the losses? I'd argue that ANY loss of crop is significant]

40 Plant -> Plants

41 involved in -> involving

53 economic loss [CITE!]

Figure 1 [MAJOR] Consider deleting; this was explained in the Introduction adequately.

Figures 2/3 [MAJOR] Nowhere in the manuscript are the frozen leaves presented for evidence of frost damage (Fig.5 is showing a different aspect of this). If freezing damage is visually visible, I'd argue for adding the leaves at treated temperatures for demonstration of that fact.

Fig.3 [MAJOR] For (B), please add the data for UFC. Also, please convert at least one datapoint into mM to enable direct comparisons of the 30 mM GB applied. Finally, which temperature treatment was assessed for 30 mM GB in (B)?

102 "reduction in injury percentages" [MAJOR] This is the first time we learn how the assays were measured (Fig.3 legend). Details of the electrolyte leakage were not described at all, although the whole study is based on them.

105 mechanism -> mechanism(s)

129 visually

Fig.6 [MAJOR] In (D), which temperature was used for the datapoint of 30 mM GB?

Fig.7 [MAJOR] Which temperature was used for the datapoint of 30 mM GB?

Fig.3,4,6, M&M description (323). [MAJOR] Be careful when stating P. In many instances in the places identified, the actual parameter should be α (type I error level).

169 its effect -> data on its effect

170 been

176 [MAJOR] This is a great spot to give a 1-2 sentence summary of the results, and whether they make you accept or reject the study hypothesis formulated in 55-56.

181-186 [MAJOR] I suggest adding this data as Supplementary: Otherwise, they do not stand a chance to be published ever, prohibiting other scholars from this potentially important insight. Also, this would allow to give justice to your hard work  on this research.

183 while -> whereas

190 These studies support... Please revert the sentence order:
Our results of 5% bigger leaf area in GB-fed leaves than in control (Table 1) support their conclusions.

192 better GB induced leaf growth -> it

193 to have -> had relatively

201 glycine betaine -> GB

202 vs. 203 osmolality vs. osmolarity. Please make sure those technical terms are applied properly, respectively.

207 statistically significant -> significantly

227 less hydrated though marginally -> marginally less hydrated

232 [MAJOR] This is a great spot to discuss, how your selected test temperatures compare with the agricultural conditions in Korea in the regions where cabbage is grown.

235 It has been well known that

252 excess -> excessive

232 be manifested -> manifest

254 tissues[1,5] (add space)

268 such -> transcriptomic activity of these; were -> was

277 higher -> had higher

296-299 [MAJOR] Please state clearly whether only a single GB application was used, vs. the 3d interval watering over the 7d.

304 vs. 312 [MAJOR] Details on the leaves used for study: Were the leaves throughout the first true petiolate leaves?

333 (and many others) Please change g for g throughout for centrifugation.

354 Please cite the source for the MDA formula.

354 dry weight -> DW

389 mixed -> mixing

399 per treatment, unless otherwise stated (several analyses used different setup, as per your descriptions).

409 conduct -> analyze

Author Response

Dear Mr. Reviewer #2,

We truly appreciate you for taking your time to handle our manuscript. We believe that the comments raised by the reviewer #2 significantly enhanced the quality of our study. In addition, we would like to thank positive evaluation on our manuscript. All valuable comments have been fully addressed in our revision. Please refer the uploaded file (Review Response (#2)-Plants-1451053.docx).

Sincerely,

Sang-Ryong Lee

Reviewer 3 Report

The manuscript describes the GB exogenous application in improving the freezing tolerance of cabbage. The output is that GB in optimal concentration could mitigate stress injury and indirectly increase the yield. The idea is interesting but I found a major problem in the data interpretation. In the introduction, part cold stress and freezing effect on the plant cell should be explained at a higher scientific level. A statistic is questionable. For instance, the authors used Student t-test and LSD (one-way ANOVA or factorial analysis of variance?). Please, provide the interactions between variables. Authors should check data analysis (for CAT and APX activity) again. The correlation analyses will definitely improve the manuscript. In my opinion, the Discussion should be also improved; there are too many repetitions of the same things. For, instance the osmoregulatory role of GB was repeated several times.

All other comments are in the text.

Author Response

Dear Mr. Reviewer #3,

We truly appreciate you for taking your time to handle our manuscript. We believe that the comments raised by the reviewer #3 significantly enhanced the quality of our study. In addition, we would like to thank positive evaluation on our manuscript. All valuable comments have been fully addressed in our revision. Please refer the uploaded file (Review Response (#3)-Plants-1451053.docx).

Sincerely,

Sang-Ryong Lee

Round 2

Reviewer 2 Report

Dear Authors,

Your revised report on GB application on cabbage leaves and analyses of such treatment impacting FT was extended for my peer-review. I acknowledge the improvements you introduced following the previous round of review.

At this stage I have fewer points to consider as possible improvements than last round, but I identified two major concerns I'd like you to ponder/justify. A handful of minor editorial suggestions is also included below, listed by their line number.

17 as [of] alondialdehyde (MDA) concentration

34 being [at]

35 [a] decrease

39 tissues exhibits

42 perturbation[s]

44 one of -> among

46 region[s]

47 [From that perspective, ]cold acclimation ...

48 involved in -> involving

49 hormone balance {Do you mean perturbed hormone homeostasis / hormone imbalance?}

52 which {Imprecision; what is meant here - the GB or the osmotic adjustment?}

56 its {Imprecision; what is meant here - the GB or the ordered state of membranes?}

Table 1.
Water content is redundant if DW/FW is also presented; those values will add up to 100% or unity. Water content preferable, for more precise number format.

116 levels based on LT50 {Repetition}

118-119 [MAJOR] You used clearly defined scale for GB concentration assessment; this can be easily converted into numerical values of native and GB+ molar concentration, to put in perspective how the treatment of 30 mM relates to the native state.

127 (B) {Please state for clarity, that this was done before the temperature treatment was applied.}

Figure 2 [MAJOR] This visual can be enhanced by pixel filtering and counting using ImageJ, for a tangible, numerical representation of the differences.

159,160 control -> in UFCs

167 Figure 6 (D) {Please state for clarity, that this was done before the temperature treatment was applied.}

176 Figure 7 {Please state for clarity, that this was done before the temperature treatment was applied.}

212 The better Improvement in GB

213 GB [the] stabilizing effect [of GB] on the 

213, 214 {What are "oxygen-evolving" activity or complexes? unclear what is meant here.}

216 higher [levels of] photo-assimilates

230 small but

231 compared to than

257 ) but milder than [of the]

263 [to] various

298 [the] Ca2+

304 more [the] freeze-tolerant

305 compared to [than]

336 following [after]

370 {Please correct throughout the "g" [weight unit] for "g" [acceleration unit]. You claimed it was done.}

398 were [visualized using] stained 

399 and [or] 3,3'

408 activity -> activities

409 was -> were

415-425 [MAJOR] Were those assays performed using the same amount of isolated protein? Otherwise, without protein concentration assessment, these values, that constitute a major part of this study, cannot be trusted at all.

430 [MAJOR] Was the ratio of sample:buffer w/v kept constant? Otherwise, these values, that constitute a major part of this study, cannot be trusted at all.

453 [MAJOR] Student t-test takes as one of the parameters the confidence level (alpha). P is the significance of the outcome of this or other statistical tests.

469 [comparative] gene expression [analyses] such as cold induced genes of genes responsible for proline biosynthesis {Too detailed info for Conclusions; if at all, should have been explored in the Discussion.}

Author Response

Dear Mr. Reviewer #2,

We truly appreciate you for taking your time to handle our manuscript. We believe that the comments raised by the reviewer #2 significantly enhance the quality of our study. In addition, we would like to thank positive evaluation on our manuscript. All valuable comments have been fully addressed in our revision. Please refer the Table below.

Sincerely,

Sang-Ryong Lee

Reviewer 3 Report

The authors improved the manuscript accordingly to the most comments. I think the correlation analyses would greatly improve data analysis. The authors mentioned that correlation analyses would affect data reconstruction and this is the reason why they did perform it.  No, it only can improve it without interfering with other stat analyze. The more stat tests you used, the better view you have. You should compare treatments with all measurable parameters by Pearson or correlation matrix.

In my opinion, the answer is not convincing enough to consider this paper for publication.

Author Response

Dear Mr. Reviewer #3,

We truly appreciate you for taking your time to handle our manuscript. We believe that the comments raised by the reviewer #3 significantly enhance the quality of our study. In addition, we would like to thank positive evaluation on our manuscript. All valuable comments have been fully addressed in our revision. Please refer the Table below.

Sincerely,

Sang-Ryong Lee

Round 3

Reviewer 2 Report

Dear Authors,

Your revised report on GB application was extended to my peer-review. In this further-improved version, I have only a handful of comments which need improvements, in my opinion.

l.38 "mostly explored" is imprecise. Do you mean "predominantly explored", "most frequently explored", "most in-depth explored", or any other?

Figure 8. [MAJOR] Please keep the same order of parameters to be compared across {B, C, and D}, for easier spotting. In relation to this, your Discussion only briefly mentions points that this Figure suggests and could be extended. For instance, one would expect mostly comparable reactions in C and D, whereas only one pairwise correlation agrees, based on their significance. [MAJOR] Please note in the Figure legend the meanings of significance indices. [MAJOR] The M&M section is missing the details on how the significance of those correlations was tested.

l.327-240 [MAJOR] Nowhere in the current version do you disclose the fact of unreliability of your enzymic assays due to lack of standardizing the results by protein concentration. I understand that protein concentration was variably used in this aspect, but it is a current standard, which your study does not meet. At the very least I expect a strong disclaimer in the Discussion to that effect.

l.431 You again falsely assured that all "g"s relating to acceleration were converted to "g"s. If such a little detail cannot be followed through, one may wonder about the quality of your research.

l.437-438 Please append to what uM concentration is 350 ug/mL equivalent to.

l.488-494 [MAJOR] In what volume was each frozen tissue sample ground? Was that ratio kept constant throughout? [MAJOR] Please append this (constant) information throughout, wherever applies. If that was not observed, I expect more disclaimers to that fact in the Discussion.

Author Response

Dear Reviewer #2,

We truly appreciate you for taking your time to handle our manuscript. We believe that the comments raised by reviewer #2 significantly enhance the quality of our study. In addition, we would like to thank positive evaluation on our manuscript. All valuable comments have been fully addressed in our revision. Please refer the attached file.

Sincerely,

Sang-Ryong Lee

Reviewer 3 Report

Dear Authors and Editor,

The manuscript is now improved by correlation analyze which contributes in explaining the protective GB role under cold stress. I found some minor mistakes regarding this analysis (pages 7 and 10) which should be fixed before publication.

Dear Authors and Editor,

The manuscript is now improved by correlation analyze which contributes in explaining the protective GB role under cold stress. I found some minor mistakes regarding this analysis (pages 7 and 10) which should be fixed before publication.

Dear Authors and Editor,

The manuscript is now improved by correlation analyze which contributes in explaining the protective GB role under cold stress. I found some minor mistakes regarding this analysis (pages 7 and 10) which should be fixed before publication.

Dear Authors and Editor,

The manuscript is now improved by correlation analyze which contributes in explaining the protective GB role under cold stress. I found some minor mistakes regarding this analysis (pages 7 and 10) which should be fixed before publication.

Dear Authors and Editor,

The manuscript is now improved by correlation analyze which contributes in explaining the protective GB role under cold stress. I found some minor mistakes regarding this analysis (pages 7 and 10) which should be fixed before publication.

Dear Authors and Editor,

The manuscript is now improved by correlation analyze which contributes in explaining the protective GB role under cold stress. I found some minor mistakes regarding this analysis (pages 7 and 10) which should be fixed before publication.

Dear Authors and Editor,

The manuscript is now improved by correlation analyze which contributes in explaining the protective GB role under cold stress. I found some minor mistakes regarding this analysis (pages 7 and 10) which should be fixed before publication.

Dear Authors and Editor,

The manuscript is now improved by correlation analyze which contributes in explaining the protective GB role under cold stress. I found some minor mistakes regarding this analysis (pages 7 and 10) which should be fixed before publication.

Dear Authors and Editor,

The manuscript is now improved by correlation analyze which contributes in explaining the protective GB role under cold stress. I found some minor mistakes regarding this analysis (pages 7 and 10) which should be fixed before publication.

Author Response

Dear Reviewer #3,

We truly appreciate you for taking your time to handle our manuscript. We believe that the comments raised by the reviewer #3 significantly enhance the quality of our study. In addition, we would like to thank positive evaluation on our manuscript. All valuable comments have been fully addressed in our revision. Please refer the Table below.

Sincerely,

Sang-Ryong Lee

Round 4

Reviewer 2 Report

Dear Authors,

Thank you for incorporation of the suggested items to ensure your submission's scientific correctness. At this stage, I only have minor suggestions, and one major point about the Fig.8. After these are satisfactorily addressed, I feel your report is  suitable for publication.

l.36 driving -> drive

39 as follows:

69 freeze thaw [related deterioration]

72 proline [content] analysis

111 Befre "Minimum..." is a perfect spot to explain that your FT injury was assessed using EL: Currently we do not know this until we get to methods part, at the very end of the MS.

155 & -> and

162 control [Here and many other places; I suggest to avoid confusion - as there are two types of controls - to either specify this in all such places as "non GB-fed control" or similar, or introduce an abbreviation, analogous to UFC. (Results, Discussion, M&M text, and Figure/Tables + legends).

174 "more repressed" suggests you know the mechanisms behind; I suggest caution and replacing with "lower", which is simpler and more neutral.

Figure 8. [MAJOR] Are those correlations for "control" or GB-fed leaves data? It might be more insightful, to present side-by-side "control" and GB-fed for even more insight.

206 IL: what does this abbreviation signify? If another term was used before, please use it here as well.

214 between - among

219-225 [MAJOR] See Fig.8 comment

233 causes lower -> mitigates

289 condition -> conditions

327 levels that are -> levels:

361 have been known to be -> are

381 Ca -> the Ca

393 its -> such a

443 first[ly]

447 made -> plotted

450 temperature[s]; first

456 heated -> heating

489 exhibited in the present study -> presented

491 enzyme activity -> enzymes activities

Author Response

Ms. Ref. No.: Plants-1451053

Title: Exogenous glycine betaine application improves freezing tolerance of cabbage (Brassica oleracea L.) leaves

Dear Reviewer #2,

We truly appreciate you for taking your time to handle our manuscript. We believe that the comments raised by the reviewer #2 significantly enhanced the quality of our study. In addition, we would like to thank positive evaluation on our manuscript. All valuable comments have been fully addressed in our revision. Please refer the attached pdf file. Thank you so much.

Sincerely,

Sang-Ryong Lee
